# Bright Fluorescent Vacuolar Marker Lines Allow Vacuolar Tracing Across Multiple Tissues and Stress Conditions in Rice

**DOI:** 10.3390/ijms21124203

**Published:** 2020-06-12

**Authors:** Yiran Cao, Wenguo Cai, Xiaofei Chen, Mingjiao Chen, Jianjun Chu, Wanqi Liang, Staffan Persson, Zengyu Liu, Dabing Zhang

**Affiliations:** 1Joint International Research Laboratory of Metabolic and Developmental Sciences, Shanghai Jiao Tong University-University of Adelaide Joint Centre for Agriculture and Health, School of Life Sciences and Biotechnology, Shanghai Jiao Tong University, Shanghai 200240, China; caoyiran@sjtu.edu.cn (Y.C.); wenguo.cai@sjtu.edu.cn (W.C.); chenxf8663@163.com (X.C.); chenmjiao@126.com (M.C.); shanghaichu@sjtu.edu.cn (J.C.); wqliang@sjtu.edu.cn (W.L.); staffan.persson@unimelb.edu.au (S.P.); 2Flow Station of Post-doctoral Scientific Research, School of Life Sciences and Biotechnology, Shanghai Jiao Tong University, Shanghai 200240, China; 3School of Biosciences, University of Melbourne, Parkville Victoria 3010, Melbourne, Australia; 4School of Agriculture, Food, and Wine, University of Adelaide, Waite Campus, Urrbrae, South Australia 5064, Australia

**Keywords:** *Oryza sativa*, vacuole, fluorescent marker, tonoplast intrinsic protein

## Abstract

The vacuole is indispensable for cells to maintain their water potential and to respond to environmental changes. Nevertheless, investigations of vacuole morphology and its functions have been limited to *Arabidopsis thaliana* with few studies in the model crop rice (*Oryza sativa*). Here, we report the establishment of bright rice vacuole fluorescent reporter systems using OsTIP1;1, a tonoplast water channel protein, fused to either an enhanced green fluorescent protein or an mCherry red fluorescent protein. We used the corresponding transgenic rice lines to trace the vacuole morphology in roots, leaves, anthers, and pollen grains. Notably, we observed dynamic changes in vacuole morphologies in pollen and root epidermis that corresponded to their developmental states as well as vacuole shape alterations in response to abiotic stresses. Our results indicate that the application of our vacuole markers may aid in understanding rice vacuole function and structure across different tissues and environmental conditions in rice.

## 1. Introduction

The vacuole is an indispensable organelle for plant cells. Because of its large size (up to 90% of cell volume), the vacuole works as a reservoir, not only for water but also for proteins, metabolites, and ions [1,2]. These characteristics provide turgor to maintain cell shape and enables chemical recycling. There are different types of vacuoles in plant cells including the central vacuole (also called lytic vacuole), protein storage vacuoles, and some pre-vacuolar compartments [2]. These vacuolar types are dynamically forming and changing in coordination with developmental and environmental cues [2,3].

A range of different tools have been utilized to understand how vacuoles change and function during plant growth and development. For example, electron microscopy and staining of the vacuole with different dyes have been used to directly observe the vacuole [2,4,5]. While these techniques have their own advantages and disadvantages, the use of fluorescently tagged markers for in vivo imaging of tonoplast behavior offers additional insights. Several types of proteins have been used as plant vacuolar maker such as vacuolar syntaxin-related proteins (SYPs), v-SNARE synaptobrevin proteins (also called vesicle-associated membrane proteins, VAMPs), and tonoplast intrinsic proteins (TIPs) [6,7,8,9]. Among these, the TIPs are widely used in *Arabidopsis* due to the fact of their general tonoplast labelling patterns. Most TIPs are tonoplast-located aquaporins which mainly transport water but also small solutes [8]. In *Arabidopsis*, the TIPs are expressed in a tissue and/or developmental specific manner [8,9]. Using TIP peptide-specific antibodies, some of the TIPs were found to label specific vacuole structures [10]. By contrast, the fluorescently labelled TIPs typically label most of the different types of vacuoles. This is prominent when they are overexpressed, for example, by using the strong 35S promoter which labels most vacuoles in many tissues [9].

While the vacuole function and morphology in *Arabidopsis* have been widely studied, there is only limited knowledge about the vacuole in rice that is grown in wet environments. Two studies have reported on fluorescent labelling of tonoplasts in rice; however, changes in vacuole morphology have not been addressed [11,12]. In this work, we designed and implemented fluorescent rice marker lines using a vacuole rice TIP protein. Using these lines, we studied specific vacuole structures in rice root cells and observed how the vacuole morphology changes during rice pollen development and during osmotic treatments. This work therefore provides a useful tool to investigate rice vacuole functions.

## 2. Results

### 2.1. Generation of Rice Vacuolar Marker Lines

To study the vacuole morphology and function in rice, we aimed to generate fluorescent vacuolar markers. The rice genome (*Oryza sativa* Japonica) encodes ten TIPs, divided into five sub-families named TIP1 to TIP5 (Appendix A). From these, we chose *O. sativa* tonoplast intrinsic protein 1;1 (OsTIP1;1) as a possible rice tonoplast labelling protein, as it is ubiquitously expressed across rice tissues (Appendix A). To generate a fluorescent version of TIP1;1, we fused the enhanced green fluorescent protein (eGFP) [13] or the red fluorescent protein mCherry [14] at the N-terminus of OsTIP1;1 (Appendix A). Both constructs were either driven by a 35S promoter (for protoplast and tobacco epidermal leaf cell expression studies) or by a maize *ubiquitin 1* promoter (*ZmUbi1*) which is widely used in monocots [15]. Topology estimates have indicated that both the N and C termini of TIP proteins face the cytosol and that they are unlikely to interfere with the channel region [16]. Indeed, neither N-terminal nor C-terminal fusions blocked the TIPs’ function in *Arabidopsis* [9]. Therefore, the tagged OsTIP1;1 is likely to maintain its function also with the fluorescent tag.

To test whether OsTIP1;1 labels the tonoplasts, we first assessed the fluorescently tagged OsTIP1;1 with a published vacuole membrane protein in transient expression systems including epidermal tobacco leaf cells and rice protoplasts. We chose the *Arabidopsis* VAMP711 (AtVAMP711), which has frequently been adopted to label the tonoplasts [17], as a tonoplast indicator. The CaMV 35S-driven mCherry-OsTIP1;1 and GFP-AtVAMP711 were, therefore, transiently co-transformed. As shown in Figure 1, the mCherry-OsTIP1;1 signal nicely co-localized with the GFP-VAMP711 fluorescence in tobacco leaf epidermal cells. Notably, the signal not only labeled the big central vacuoles (i.e., tonoplasts running along the cell edges (Figure 1a)) but also the bubble-like small vacuoles (Figure 1b). The fluorescence of the small vacuoles was relatively brighter than the big vacuoles (the signal intensity ratio was approximately 8:1 for the small vacuoles to the large ones). These data indicate that the TIPs are populating the smaller tonoplasts with higher density than that of the larger ones. Similarly, the OsTIP1;1 also labelled the tonoplast in rice protoplasts (Figure 1c).

Subsequently, we transformed rice calli with the *ZmUbi1::GFP–OsTIP1;1* and *ZmUbi1::mCherry-OsTIP1;1* plasmids by Agrobacterium infection and obtained transgenic lines that we refer to as GFP–OsTIP1;1 and mCh–OsTIP1;1 hereafter. More than 10 independent T0 lines were obtained for each transformation without any obvious plant developmental defects compared to the control (Appendix A). We grew these lines for three generations and observed clear fluorescent signals among plants in all generations. These signals were not seen in non-transformed wild-type plants under the same imaging settings (Appendix A). These results indicate that our OsTIP1;1 lines may be used as stable marker lines in rice. In this article, we mainly show the data from the GFP–OsTIP1;1 lines.

### 2.2. GFP–OsTIP1;1 Fluorescence Revealed Diverging Vacuolar Morphology Among Rice Tissues

Using the stable transgenic GFP–OsTIP1;1 lines, we checked the fluorescence in different rice tissues. Clear signals were observed in all investigated rice tissues with tonoplasts clearly labelled. We found that the vacuole morphology in different rice tissues varied. As shown in Figure 2, coleoptile cells were full of small bubble-like vacuoles (Figure 2a,b), while the vacuoles in true leaves, both pavement and vascular cells, were occupied by big central vacuoles (Figure 2d,e). The central vacuoles were usually not observed in the epidermis of coleoptiles, and small bubble-like vacuoles were only occasionally observed in the true leaf epidermal cells. If the expanded central vacuoles supported the expansion of true leaf cells, the folded coleoptile shape was probably the result of fragmented vacuoles with low turgor. By contrast, the cells of the flat *Arabidopsis* leaves were typically occupied by big central vacuoles, independently if in the cotyledon or true leaf, in young or mature leaves [9].

At the leaf surface, the stomata in the coleoptile are dicot-like with only two guard cells (Figure 2c), while they are monocot-like in true leaves with both guard and subsidiary cells (Figure 2f–i). In guard cells of coleoptiles, we typically observed fragmented but expanded vacuoles which corresponded to open stomata. This vacuolar configuration was similar to those reported in *Arabidopsis* stomata where the vacuole sizes decreased during stomata closure [6,18,19]. In rice true leaves, the large central vacuole was rarely seen in the subsidiary cell. As shown in Figure 2f–i, one relatively large as well as some smaller vacuoles typically occupied each subsidiary cell. Fused tubule-like vacuoles occupied most of the cell volume in guard cells associated with open stomata (Figure 2h), while highly fragmented vacuoles were evident in guard cells associated with fully closed stomata (Figure 2f). The vacuole discrepancy in these two types of stoma might indicate that the vacuole shapes change according to the stomata status. Hence, this observation is consistent with a previous observation that the vacuole size drives guard cell shape and thus stomata function [19].

We further studied the variety of vacuolar morphology in other tissues such as rice reproductive organs (Appendix A). The vacuoles in stamen filaments (Appendix A) were small but crowded, similar to those in elongated coleoptile cells (Figure 2b), perhaps indicating related vacuolar functions in these two cell types. Vacuoles in the stigma were relatively large compared to those in stamen (Appendix A). Notably, each ovary epidermal cell contained several bubble-like small vacuoles with two larger vacuoles at the ends of the cell (Appendix A). Based on these results, we propose that the various vacuole morphologies may reflect different biological processes in different cells.

### 2.3. Vacuole Morphology Changed During Rice Pollen Development

Pollen development is crucial for rice fertility and productivity. The vacuolization is an important procedure during pollen development [20,21]. Defects in rice microspore vacuolization may lead to male sterility [22,23,24]. The developmental progression is divided into 14 stages in rice pollen [25]. We followed the vacuolar morphology changes beginning at stage 8a (S8a), at which the microspores displayed dyad divided from pollen mother cell during meiosis I [25]. This stage progresses into meiosis II, where each cell of dyad continues to divide into two cells forming the tetrad. As shown in Figure 3a, the vacuole morphology in dyad and tetrad cells was similar to each other with a few small vacuoles dispersed across the cell. After degradation of the cell walls of tetrads, the microspores were released, and the vacuoles increased in number but not size (Figure 3a and Appendix A). At this stage, the vacuoles became denser but not overly crowded. After that, pollens started to increase in size but without a corresponding increase in vacuole size, leading to hollow pollens (Figure 3a and Appendix A). A lag in vacuolization occurred at late in stage 9 and ended at stage 10, during which the small vacuoles were fused into big central vacuoles. A large vacuole gradually occupied the whole microspore volume and supported the round and transparent pollen grains (Figure 3c and Appendix A). Subsequently, the central lytic vacuole was fragmented into storage vacuoles/vesicles (Figure 3c and Appendix A) that accumulate a range of chemical compounds [26]. As a consequence, the microspores collapsed and displayed irregular shapes. Henceforward, the vacuoles increased in number but decreased in size until the pollen was fully mature (Figure 3c, Appendix A). To clarify, the vacuole organization and progression during the microspore development are diagramed in Figure 3b,d.

Changes in vacuole morphology during development is a common property in rice cells. For example, invaginated vacuoles (Appendix A) were at first slightly fragmented (Appendix A) and then fused to form a big central vacuole (Appendix A) at the anther epidermis (about stage 8 to 11 of pollen development), perhaps to support anther elongation. During the rice endosperm development, proteins and starch are gradually accumulating. Based on our observations, the vacuoles were large (probably lytic vacuole) occupying the whole cell volume during early endosperm stages (Appendix A). Later, the vacuoles were gradually fragmented into smaller bubbles (Appendix A). In mature endosperms, each granule was composed of hundreds of storage vacuoles (Appendix A). This process is similar to the formation of the storage vacuole in microspores, which may represent a common course of storage vacuole formation, i.e., via a fragmentation of the big lytic vacuole. Similar vacuolar morphology alterations were also observed in developing root hairs (Appendix A).

### 2.4. Vacuolar Morphology Changes During Root Cell Development

Due to the fact of its water storing function, the vacuole is critical for rice to adapt to aquatic growth, so we closely observed the vacuole morphologies in rice root cells. Different types of roots, including radicle, crown, and lateral roots were observed. Based on the observations of epidermal cells (Appendix A), there was no significant difference among radicle and crown roots. In the root caps, under the caducous cells (i.e., detached cells), each cell usually had a big round vacuole that did not occupy the whole cell volume (Figure 4a–c). However, in the subapical regions, the vacuole morphologies diverged substantially across different developmental zones (Figure 4d,e). Both dividing and transition cells were highly compartmentalized, containing substantial numbers of tonoplast invaginations (Figure 4d,e). The compartment numbers of vacuoles were dramatically decreased in elongating and mature cells (Appendix A).

In lateral roots, we observed two types of vacuole morphologies: one was compartmentalized, similar to that in the radical and crown roots (Figure 4f), and the other was lateral root specific, characterized by an over saturation of GFP fluorescence (Figure 4g). When observing the latter more closely, some of the bright structures appeared globular (Figure 5a, white arrow heads) and some strand-like (Figure 5a, yellow arrow heads). These structures seemed to divide the cells into small compartments. From the division to elongation zone, these bright structures’ numbers decreased (Figure 5b), while the area ratio of compartments to cell increased (Figure 5c). Consequently, the total vacuole-to-cell area did not change much among the different zones (Figure 5d).

What, then, are those bright vacuole structures in lateral roots and how do they form? These structures are not penetrating the whole cell, since different structures were seen at different Z-planes (Figure 5e). Notably, the bright structures appeared to originate from the peripheral tonoplast, indicated by arrow heads in Figure 5e. Using time-lapse tracking of the vacuoles, we found that both the strands and bubbles were dynamic (moving in the vacuole lumen) and interchangeable (Figure 5f; Movie S1). The yellow triangles in Figure 5f are following the transition of a strand to a bubble. Based on these observations, we speculate that the vacuole in each cell is an intact and connected membrane system. Perhaps due to the space restrictions, the tonoplasts are invaginated in dividing and transitioning cells. When the cells elongate, the invaginated membrane expands and, therefore, the bright structures disappear.

We next assessed if the vacuole shapes coincided with changes in the vacuole pH. We therefore used the pH sensitive dye BCECF-AM (2’,7’-bis-(2-carboxyethyl)-5-(and-6)-carboxyfluorescein acetoxymethyl ester) [5,27] in roots of our transgenic mCh–OsTIP1;1 lines. In Appendix A, we show that the BCECF fluorescence was retained in the vacuole lumens, as it was surrounded by the mCh–OsTIP1;1 signal. The signal intensity of BCECF differed in different root cells, perhaps indicating that the vacuole of different cells may have different pH. However, no correlation was found between the signal intensity and the vacuole shape.

The invaginated and globularized vacuoles in rice roots are not typically seen in *Arabidopsis*. We therefore consider that the vacuole morphology may differ between *Arabidopsis* and rice root cells. To support this consideration, we took images of vacuoles in *Arabidopsis* roots, labelled by mCherry-AtVAMP711 [17]. Similar to previous reports [7,28], the vacuoles were seen as small but dense bubbles, sometimes with tubules (Appendix A). As shown in Appendix A, the mCh–OsTIP1;1 lines displayed similar vacuole structures as the GFP lines, which again supports the difference between rice and *Arabidopsis* root epidermis. Besides the morphological differences, the volume occupied by vacuoles were notably larger in rice root cells as compared to *Arabidopsis* roots.

### 2.5. Vacuoles in Rice Root Cells Change When Exposed to Abiotic Stress

Vacuoles regulate the water potential of the root cells when they encounter environmental changes. However, it is unclear how different conditions impact on the morphology and how the progression of such change occurs. We first treated the fluorescent marker lines with 18% polyethylene glycol 4000 (PEG4000). In our treatments, PEG4000 induced bubble-like vacuoles with high fluorescence intensity compared to other tonoplasts (Figure 6c,d). This result corresponds to the reported PEG8000 treatment of tobacco suspension cells, where small bright vacuoles were induced inside the lytic vacuoles [29]. Under our condition, this phenomenon was mainly seen in cells of the transition and division zone. We rarely saw this behavior in elongating cells. In some instances, we also observed the bright bubble-like vacuoles in control root cells but with much lower frequency (Figure 6a,b). These bright structures appeared to be on the surface of the main vacuole but not inside the big vacuole as reported in the tobacco suspension cells [29]. In addition, we used 150 mM NaCl to further study potential vacuolar responses. Similar to the PEG treatment, the salt treatment also caused TIP signal accumulation (Figure 6e–h). Notably, here, the accumulation was not uniform, but instead formed high intensity segments on the tonoplasts. The white arrows in Figure 6g,h indicate these segments. The effect was seen in all root cell types (Appendix A). Again, this non-uniform accumulation was also occasionally observed in controlled roots, which might be due to some mild and unintended stress.

## 3. Discussion

### 3.1. The Vacuole Morphology Changes During Development

Vacuoles regulate the cell water potential and determine the shape of cells. Hence, the vacuoles adapt to the demands of the cell and to the progression in cell size and function. In this study, we followed changes in the vacuole morphologies during the development of different cell types, including those in roots, pollen, anthers, endosperm, and root hairs. The pollen is an interesting organ to study the vacuole progression, since the pollen itself undergoes a series of morphological changes during maturation [30,31,32]. We found that the vacuole numbers and size changed during pollen development (Figure 3). In this process, the transition from the lytic vacuole to potential protein storage vacuoles appears to be an important landmark. Here, the lytic vacuole was first fragmented and then perhaps reprogrammed into small storage vacuoles (Figure 3c). This course of vacuole changes was also observed during rice endosperm formation (Figure 4d–f). So, we hypothesize that the lytic vacuole first is split and is then reprogrammed to obtain a different function during development. Such changes appear to be a common theme in rice cells. These observations are consistent with the result in *Arabidopsis* embryos, where pre-existing vacuoles are reprogrammed to form the protein storage vacuole in developing cotyledons [33]. The other way around may also occur, i.e., the protein storage vacuoles become reprogrammed to form new lytic vacuoles, for example, in *Arabidopsis* root meristems [34]. Therefore, the vacuole shape and function are interchangeable, which may be an efficient way to respond to different needs of the plant cell.

### 3.2. Vacuolar Responses to Environmental Cues

Vacuoles are thought to respond to environmental cues. In our conditions, both PEG4000 and NaCl treatments induced local fluorescence intensity changes, displayed as segments along the tonoplasts or as small bubbles (Figure 6 and Appendix A). This phenomenon was, however, also occasionally found in the untreated cells. Osmotic stresses typically increase gene expression of tonoplast aquaporins [35,36]. These results may indicate that plants under such stresses demand more aquaporins to regulate water and solute contents. However, why the GFP–OsTIP1;1 fluorescence displayed non-uniform patterns is not clear. We can imagine two possibilities: one is that the loading capacities and demands along the tonoplast are not equal; and the other is that stresses can induce local membrane folding. Nevertheless, these non-uniform distribution patterns of GFP–OsTIP1;1 need to be further investigated to better understand local variations along the tonoplasts.

### 3.3. Vacuole Diversity Among Plant Species

In this study, we compared the vacuole morphologies and organizations between rice and *Arabidopsis* root cells. In rice root cells, the tonoplasts are highly invaginated to form many compartments and perhaps to increase the tonoplast membrane surface (Appendix A). While the vacuoles in *Arabidopsis* root cells displayed many connected bubbles, the vacuole occupancy in rice root cells were notably higher than that in *Arabidopsis*. Whether these changes in membrane surface area enhance the ability of cells to better withstand environmental changes are not known. But, given the different growth environments between rice and *Arabidopsis*, such abilities may perhaps be expected. Nevertheless, it is clear that the vacuole organization differs between plant species. Similar divergences in vacuole morphology are evident in pollen grains. During rice pollen development, only one vacuolization occurred, just before mitosis during which the vacuole occupies almost the whole microspore volume (Figure 3). This phenomenon is similar to *Lolium perenne* [37]. Several other species, including *Arabidopsis*, also display only one vacuolization, but *Lycopersicum peruvianum* and *Olea* were found to have two [26]. By contrast, no vacuolization event was reported in Orchids in which small vacuoles are maintained throughout the microspore development. Such small vacuoles or vesicles that we observed in mature rice pollen have been observed also in mature pollen of *Arabidopsis* and Orchids [38,39]. However, vacuoles were absent from mature pollen of many other plant species. Whether these differences are due to the environmental constraints or if they are of developmental relevance remain to be investigated.

To conclude, we have generated bright fluorescent vacuole marker lines to explore vacuole morphology during development and in response to environmental changes.

## 4. Materials and Methods

### 4.1. Plasmids Construction

To generate the transgenic rice line, the pCAMBIA 1301 vector was used. The *mazie ubiquitin1* promotor (1981 bp upstream of the start codon) was inserted into the vector at EcoRI site. After that, *eGFP* was inserted downstream of the promoter at KpnI site. Lastly, the *OsTIP1;1* (Os03g0146100) CDS (length: 750 bp) was amplified from *Oryza sativa* Japonica cDNA and fused in frame with eGFP by infusion between HindIII and PmlI. To generate *Ubi1: mCherry-TIP1;1* plasmid, *eGFP* was replaced with *mCherry*. The plasmids were transformed into rice calli by Agro-infiltration. Transgenic plants were obtained through several rounds of differentiation inductions, selected by the hygromycin B medium.

To perform the transient expression experiments, the *mCherry-TIP1;1* was driven by 35S promoter in the vector of pCAMBIA 1301, inserted between sites of SpeI and BstEII. The *AtVAMP711* CDS (AT4G32150, length: 1363 bp) was amplified from *Arabidopsis* cDNA library and inserted into a modified vector of pXY104 [40] with the nYFP cassette removed. The e*GFP* was inserted at XbaI site downstream of the enhanced 35S promoter and *AtVAMP711* was inserted at PstI site.

### 4.2. Tobacco Transformation

To transiently express proteins in tobacco leaves, the method from Li [41] was adopted with minor modifications. Before the infiltration, the agrobacterium containing plasmid of *35s:mCh–OsTIP1;1* were adjusted to OD600 of 1. Moreover, the agrobacterium containing plasmid of *35s:GFP-AtVAMP711* were adjusted to OD600 of 0.5. Then the two were mixed (one to one volume) and infiltrated into 6 weeks old tobacco leaves. After incubation in the dark for 40–48 h, leaf pieces were observed under confocal microscope.

### 4.3. Protoplast Transformation

The rice protoplast extraction and transformation were improved from the *Arabidopsis* method [42]. The shoots of one-week-old dark-grown rice seedlings were cut into pieces, as small as possible. These pieces were digested with 20 mL enzyme solution (1% cellulase and 0.25% macerozyme, Yakult, Tokyo, Japan; 10% albumin bovine, Amresco, Framingham, MA U.S.) for 2–4 h in dark after vacuum infiltration for 10 min. After the digestion, the protoplasts were filtered by a 70 µm nylon mesh, followed by spinning to collect the protoplasts. Protoplasts were resuspended and washed with W5 solution (150 mM NaCl, 10 mM CaCl_2_, 5 mM KCl, 2 mM MES, pH = 5.7; Sangon Biotech, Shanghai, China). Lastly, 1~2 mL MMg solution (400 mM mannitol, 15 mM MgCl_2_·6H_2_O, 4 mM MES, pH = 5.7; Sangon Biotech, Shanghai, China) was added to resuspend the protoplasts. Protoplast quality was checked by optical microscope and counted with a hemocytometer (20~25 cells/unit). For each transformation, 100 µL quality checked protoplasts and 10 µg plasmid DNA were used. After the PEG-mediated transformation, the protoplasts were washed and incubated at 28 °C for 24 h, after which fluorescent signals were observed by confocal microscope.

### 4.4. Plant Growth Condition

At the seedling stage (1-4 weeks), the rice plant (*Oryza sativa Japonica*, cultivar 9522) were grown in 1/2 MS liquid medium in a Percival with the conditions of 28 °C/23 °C, 16 h light/8 h dark. Large plants were transferred to growth chamber with similar conditions, cultured in liquid media supplied with nutrients, where the plants would flower and set seeds.

### 4.5. PEG and Salt Treatment

We used 2 week old transgenic plants to do the treatments. Intact roots, without detaching from the plants, were immersed into either control medium (1/2 MS liquid medium) or the control medium plus PEG4000 (A620431-0500, Sangon Biotech, Shanghai, China) or NaCl (A100241-0500, Sangon Biotech, Shanghai, China). Approximately 1 cm length root tips were detached and imaged by confocal microscope within 30 min.

### 4.6. BCECF Staining

This method was modified from Scheuring et al. [5]. One-week-old mCh–OsTIP1;1 seedlings were stained. The lateral roots were immersed into staining solution containing 10 µM BCECF-AM (S1006, Beyotime Biotechnology, Shanghai, China), 0.02% hypochlorous acid, and 1/2 MS. After incubation in the dark for approximately 15 min, the roots were washed gently and observed under a confocal microscope.

### 4.7. Confocal Microscope Observation

A Leica SP5 laser scanning confocal (Leica, Weztlar, Germany) was used to collect the fluorescence images. To observe the signals in different tissues, we removed small pieces (less than approximately 0.5 cm^2^) from tissues and put them on glass slides, mounted with water. The observation for each slide was less than 30 min unless for time-lapse imaging. All the images were obtained from the epidermis if there was no other specific indication. Before imaging our sample, wild-type plants were checked to make sure auto-fluorescence was not interfering with our imaging. To label the cell boundary in root, the detached root tips were stained with propidium iodide (PI, P4170-100 mg; Sigma, Cleveland, OH, USA) for 10 min in the dark. For the z-stack continuous images, the step size was set as 0.5 µm. For the time-lapse images, the interval time was set at 10 s. The GFP excitation wavelength was 488 nm, and the gain wavelength was 500–550 nm. The mCherry excitation wavelength was 561 nm, and the gain wavelength was 580–630 nm. The PI signal was excited at 561 nm and recorded between 600 and 680 nm. The BCECF was excited at 561 nm and recorded between 580 and 650 nm. To reduce interference between channels, we used a sequential method to capture the BCECF and mCherry signals together.

## Figures and Tables

**Figure 1 ijms-21-04203-f001:**
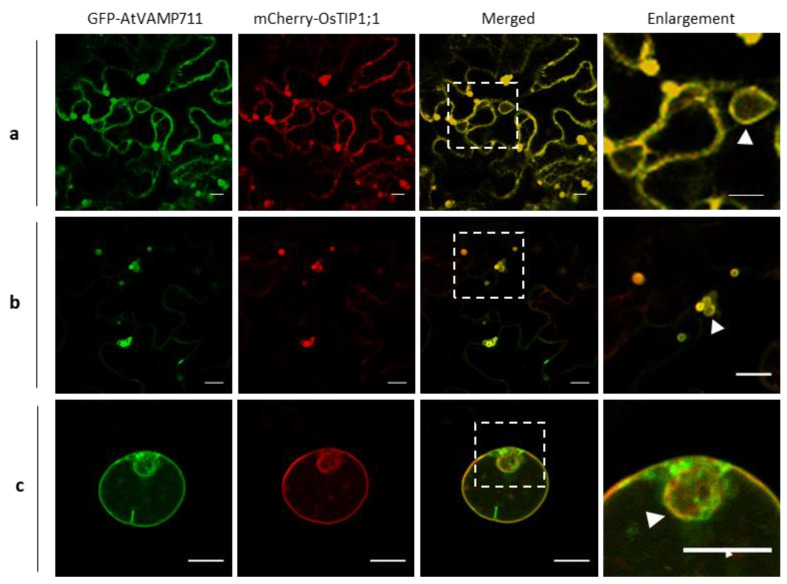
Co-localization of OsTIP1;1 (tonoplast intrinsic proteins 1;1 in rice) and AtVAMP711 (vesicle-associated membrane protein 711 in *Arabidopsis*) in transient expression systems. (**a** and **b**) Transient expression in epidermal tobacco leaves. OsTIP1;1 co-localized with tonoplast protein AtVAMP711 at central (lytic) vacuoles (**a**) and small bubble-like vacuoles (**b**), respectively. The laser intensity of the GFP (green fluorescent protein) channel was 10% in (**a**) and 5% in (**b**), and other settings including the emission, smart gain, and offset were the same. (**c**) Co-localization between OsTIP1;1 and AtVAMP711 in rice protoplast. The dashed white squares indicate regions of enlargement (right panel). The arrow heads indicate presumed vacuoles. Scale bars: 10 µm.

**Figure 2 ijms-21-04203-f002:**
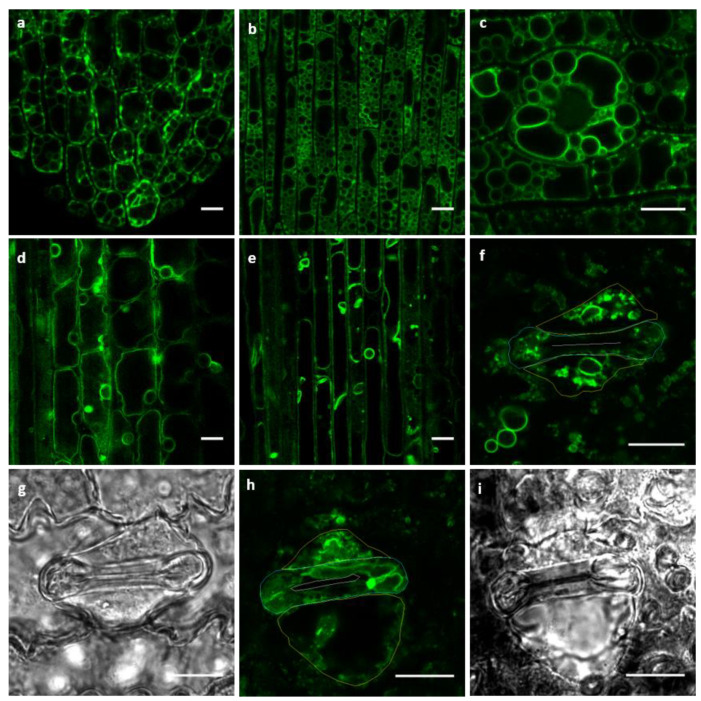
Vacuole morphology differences across multiple rice leaf cells. (**a**,**b**) Coleoptile epidermal cells are enriched with small bubble-like vacuoles. Coleoptiles, five days post-germination, were used to take images from the tip (**a**) and middle (**b**) regions. (**c**) The stomata in the coleoptile was packed with fragmented vacuoles. (**d,e**) Central vacuoles in true leaves. The big central vacuoles occupied most of the leaf pavement cells (**d**) and developing veins (**e**). (**f**) Vacuoles in closed stomata of true leaf. (**g**) The corresponding bright field channel of (**f)**. (**h**) Vacuoles in opened stomata of true leaf. (**i**) The corresponding bright field channel of (**h)**. Yellow and blue lines indicate the cell boundaries of subsidiary cells and guard cells, respectively. White lines indicate the pore region of the stomata. Scale bars: 10 µm.

**Figure 3 ijms-21-04203-f003:**
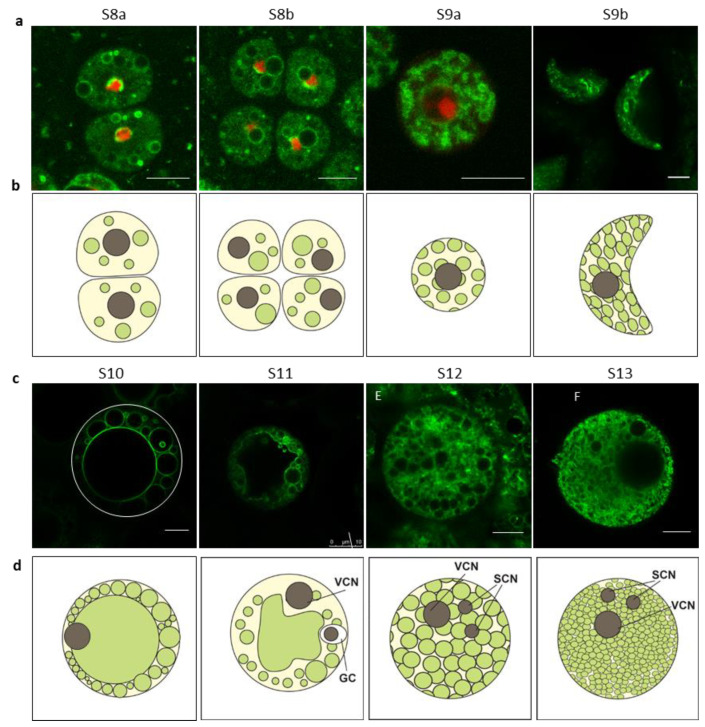
Vacuole morphology changes during microspore development. (**a,c**) Vacuole morphology progression, observed by confocal microscope, during different microspore developmental stages. Scale bars: 10 µm. White line enclosure indicates the cell edge (in (**c**); left panel). (**b,d**) Schematic views of the vacuole progression in microspores based on multiple confocal observations (*n* > 3). The shapes filled with green represent the vacuoles. VCN: vegetative cell nucleus. GC: generative cell. SCN: sperm cell nucleus. The scale is relative (**b**,**d**).

**Figure 4 ijms-21-04203-f004:**
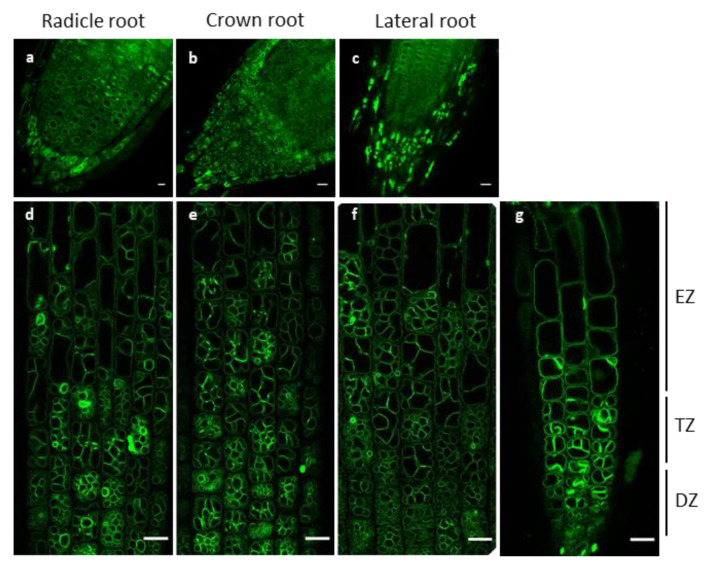
Different root types display similar vacuole organizations in their epidermal cell layers. (**a**–**c**) The root cap regions of the radical, crown, and lateral roots, respectively. (**d**–**f**) The subapical region of radical, crown, and lateral roots, respectively. (**g**) The subapical region of a lateral root which shows a different vacuole morphology compared to other root types. Approximate sub-apical regions are indicated as elongation zone (EZ), transition zone (TZ), and division zone (DZ). Scale bars: 10 µm.

**Figure 5 ijms-21-04203-f005:**
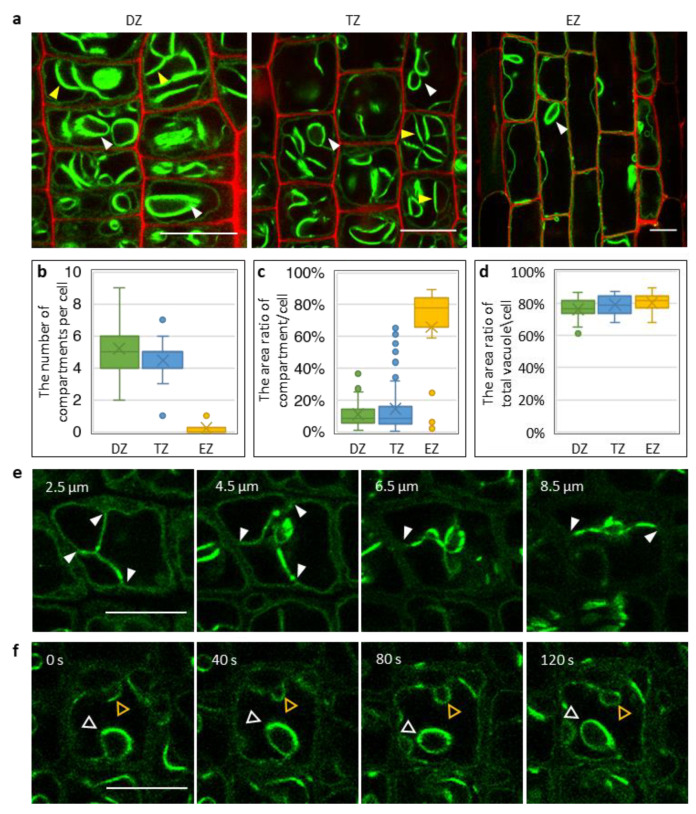
Vacuoles in rice lateral root epidermis are continuous structures with multiple invaginations. (**a**) Vacuole (in green) morphology in different lateral root zones. The cell boundary was stained with propidium iodide (in red). White triangles indicate bubble-like structures, while the yellow triangles indicate strand-like structures. Both are thought to be formed by tonoplast invaginations. (**b**) The number of invaginations in each cell decreased from DZ to EZ. (**c**) The area ratio between segmented vacuoles to cell surface/boundary. The area ratio increased from DZ to EZ. (**d**) The area ratio of total vacuole to cell surface. Further explanation to these measurements is indicated in Appendix A. Each box is the result of more than 15 cells in three roots. (**e**) Vacuole morphology via z-stacks by confocal microscope with step size of 0.5 µm. The numbers indicate the distance below the cell surface. The arrow heads indicate invagination sites at the boundary membrane. (**f**) Time-lapse images tracking the dynamics of the vacuole invaginations. The orange triangles follow a change from a strand-like structure to a bubble-like structure. The white triangles track the movement of a bubble-like structure. Scale bars: 10 µm.

**Figure 6 ijms-21-04203-f006:**
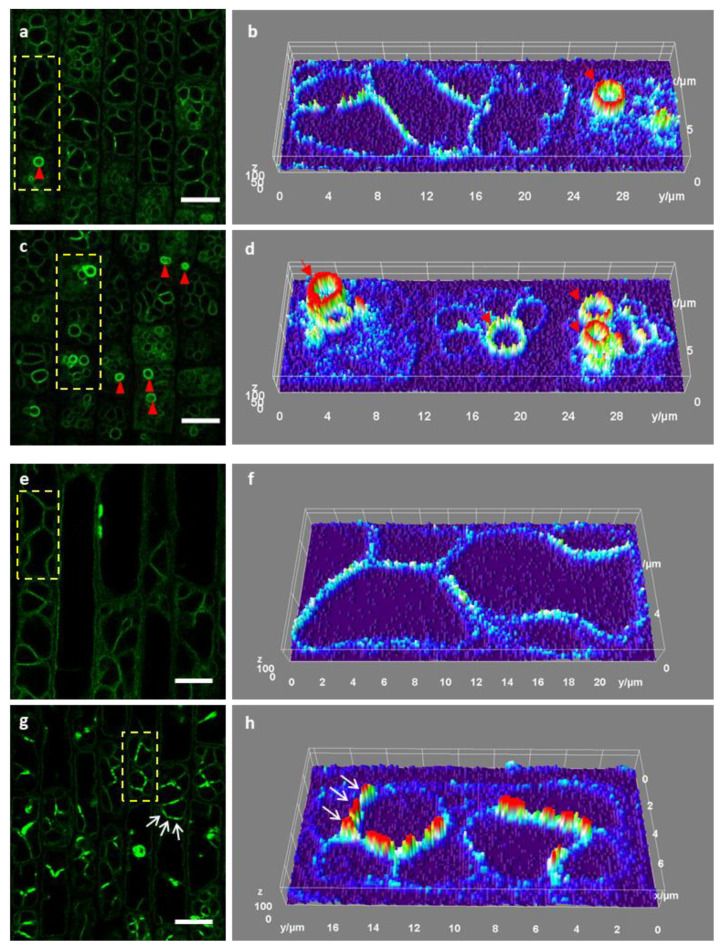
Vacuolar response to abiotic treatments in rice roots. (**a**) The vacuole morphology in control rice root cells of the transition zone. (**b**) 3D surface plot of signal intensity from the yellow, dashed rectangle in (**a**). (**c**) The vacuole morphology in rice root cells of the transition zone treated with 18% PEG (polyethylene glycol 4000) for 30 min. The red triangles (in **a** and **c**) indicate bright bubble-like vacuoles and correspond to the red arrows (in **b** and **d**). (**d**) 3D surface plot of signal intensity from the yellow, dashed rectangle in (**c**). (**e**) The vacuole morphology in control rice root cells of the elongation zone. (**f**) 3D surface plot of signal intensity from the yellow, dashed rectangle in (**e**). (**g**) The vacuole morphology in rice root cells of the elongation zone treated with 150 mM NaCl for 50 min. (**h**) 3D surface plot of signal intensity from the yellow, dashed rectangle in (**g**). The white arrows (in **g** and **h**) indicate the vacuoles with non-uniform signal distribution. Scale bars: 10 µm.

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
