# Peer review of "Bright Fluorescent Vacuolar Marker Lines Allow Vacuolar Tracing Across Multiple Tissues and Stress Conditions in Rice"

_ijms, 2020, doi:10.3390/ijms21124203_

Round 1
Reviewer 1 Report
The manuscript by Cao et al. reports on the generation of transgenic rice lines expressing a tonoplast intrinsic protein fused to either the green or mCherry red fluorescent proteins. Then, they move on to use these lines to look at dynamic changes in vacuole morphology in different rice tissues and in roots under salt or osmotic stress. Although the lines generated by Cao et al. are potentially very important for studies about monocot vacuoles, in its present form the manuscript seems to report preliminary data, and its value would be greatly enhanced if further experimentation were added. For example, I wonder if fluorescent dyes might be used in conjunction with TIP-GFP, for example, to detect acidity of the structures labeled by TIP-GFP, their hydrolytic activity (if any), and so on, which might greatly help to go beyond the mere morphological description of the structures seen by LSCM. Likewise, the discussion might be expanded a little bit more, now it mostly repeats the idea that differences in vacuolar morphology may reflect different vacuolar functions in different tissues, which is reasonable, but not enough for place the findings of this paper into a wider context. Some specific points: A concern about GFP fluorescence: given the fact that plant tissues, particularly the epidermis, can have bright auto-fluorescence in the GFP region of the spectrum, did the authors include non-transformed blanks to rule out possible auto-fluorescence? Regarding the stomata in Fig.2c, it is impossible for me to discern whether there are two guard cells surrounding a stomatal pore (which, by the way, appears faintly green, which I would not expect). It would help to see images of chlorophyll auto-fluorescence, and I imagine that propidium iodide staining might help to delineate the plasma membrane and, therefore, outline the silhouette of both guard cells. Lines 124-135: are the authors suggesting that stomatal closure is accompanied by fragmentation of the central vacuoles? Are there any observations of open vs. closed stomata in Fig.2 to support this suggestion? Line 186: is “caducous cell” a widely used denomination? Line 225: can we be sure that Fig. 5f is showing the transition from a strand to a bubble? Can we rule out a bubble moving around the focal plane? Lines 293-294: “This nonuniform distribution of OsTIP1;1 might signify that different parts of the vacuole is in need of different loading capacities during the stresses”. I can´t make sense of this suggestion, different capacity for loading water in different parts of the same vacuole?

Reviewer 2 Report
The manuscript of Cao and co-workers holds a lot of merit when create a range of vacuolar markers lines to investigate the function of vacuole in rice. The results are well described in most section of the manucript. Howeveer, the discussion is too short. This section might be imporved to proved more interest to the readers and the real scientific contribuiton of the work. Thus, the manuscript needs another round of revision before acceptance by ijms.Author Response
Thanks a lot for the suggestions. We have added new points in the discussion (please see page 11 line 298-327).
Reviewer 3 Report
The study was focused on the establishment of a bright rice vacuole fluorescent reporter system using OsTIP1;1, a tonoplast water channel protein, fused to either an enhanced green fluorescent protein or an mCherry red fluorescent protein. The Authors used the corresponding transgenic rice lines to trace the rice vacuole morphology progression in roots, leaves, anthers and pollen grains. Dynamic changes of vacuole morphologies in pollen and root epidermis that corresponded to their developmental states, as well as morphological responses to abiotic stresses were demonstrated. Results revealed that the application of the vacuole marker may significantly aid in understanding rice vacuole function and structure across different tissues and environmental conditions.
The paper is valuable and increase contribution to the research field. However, some improvements are recommended:
- Discussion and interpretation of the results are surprisingly superficial and should be THOROUGHLY revised.
- English style and grammar should be significantly improved.
- Technical issue: Figure 5B should be presented separately in order to increase its readability.
